# Validation and Comparison of Physical Models for Soil Salinity Mapping over an Arid Landscape Using Spectral Reflectance Measurements and Landsat-OLI Data

Z. M. Al-Ali [1], A. Bannari [2,*], H. Rhinane [3], A. El-Battay [4], S. A. Shahid [5] and N. Hameid [2]

1 Department of Natural Resources and Environment, College of Graduate Studies, Arabian Gulf University, Manama 26671, Bahrain; zahraamma@agu.edu.bh
2 Department of Geoinformatics, College of Graduate Studies, Arabian Gulf University, Manama 26671, Bahrain; Nadir@agu.edu.bh
3 Faculty of Sciences Ain Chock, University Hassan II, Casablanca 20100, Morocco; HASSAN.RHINANE@univh2c.ma
4 International Center for Biosaline Agriculture (ICBA), Dubai 14660, UAE; a.elbattay@biosaline.org.ae
5 Kuwait Institute for Scientific Research, 24885, Safat 13109, Kuwait; sshahid@kisr.edu.kw
* Correspondence: abannari@agu.edu.bh; Tel.: +973-1723-9545; Fax: +973-1723-9552

**Abstract:** The present study focuses on the validation and comparison of eight different physical models for soil salinity mapping in an arid landscape using two independent Landsat-Operational Land Imager (OLI) datasets: simulated and image data. The examined and compared models were previously developed for different semi-arid and arid geographic regions around the world, i.e., Latino-America, the Middle East, North and East Africa and Asia. These models integrate different spectral bands and unlike mathematical functions in their conceptualization. To achieve the objectives of the study, four main steps were completed. For simulated data, a field survey was organized, and 100 soil samples were collected with various degrees of salinity levels. The bidirectional reflectance factor was measured above each soil sample in a goniometric laboratory using an analytical spectral device (ASD) FieldSpec-4 Hi-Res spectroradiometer. These measurements were resampled and convolved in the solar-reflective bands of the Operational Land Imager (OLI) sensor using a radiative transfer code and the relative spectral response profiles characterizing the filters of the OLI sensor. Then, they were converted in terms of the considered models. Moreover, the OLI image acquired simultaneously with the field survey was radiometrically preprocessed, and the models were implemented to derive soil salinity maps. The laboratory analyses were performed to derive electrical conductivity (EC$_{-Lab}$) from each soil sample for validation and comparison purposes. These steps were undertaken between predicted salinity (EC$_{-Predicted}$) and the measured ground truth (EC$_{-Lab}$) in the same way for simulated and image data using regression analysis ($p < 0.05$), coefficient of determination ($R^2$), and root mean square error (RMSE). Moreover, the derived maps were visually interpreted and validated by comparison with observations from the field visit, ancillary data (soil, geology, geomorphology and water table maps) and soil laboratory analyses. Regardless of data sources (simulated or image) or the validation mode, the results obtained show that the predictive models based on visible- and near-infrared (VNIR) bands and vegetation indices are inadequate for soil salinity prediction in an arid landscape due to serious signals confusion between the salt crust and soil optical properties in these spectral bands. The statistical tests revealed insignificant fits ($R^2 \leq 0.41$) with very high prediction errors (RMSE $\geq 0.65$), while the model based on the second-order polynomial function and integrating the shortwave infrared (SWIR) bands provided the results of best fit, with the field observations (EC$_{-Lab}$), yielding an $R^2$ of 0.97 and a low overall RMSE of 0.13. These findings were corroborated by visual interpretation of derived maps and their validation by comparison with the ground truthing.

**Keywords:** soil salinity; salinity models; remote sensing; spectral reflectance simulation; Landsat-OLI image; electrical conductivity; validation

## 1. Introduction

During this century, soil quality has become the center of several global issues, such as food and water security, biodiversity protection and conservation, climate change mitigation, ecosystem services and sustainable development. Unfortunately, soil salinity, or salinization, is a global environmental threat, particularly in arid lands [1]. This complex and multi-factorial phenomenon can be caused by many factors or combinations of several natural and anthropogenic factors, i.e., water stress, high temperatures and increased evapotranspiration rates [2], human activities [3], and global warming [4–6]. Moreover, natural salt accumulation processes are associated and accelerated with certain physical factors, such as soils properties and permeability, geological and geomorphological aspects, water table depths, micro-topographic conditions, water use, rainfall, and salinity of groundwater [7–10]. Soil salinity can develop in different ways in almost all geographic locations and under different climatic conditions. Around the world, the total area of salt-affected soils is about 831 million hectares (ha), which includes 397 million hectares of saline soils and 434 million hectares of sodic soils [11,12]. In arid landscapes, more than 15% of the land is affected by salinity, which has been estimated to be around 45 million ha in irrigated lands and about 32 million ha in non-irrigated lands [13]. Regrettably, it is also expected to increase by 10% per year due to climatic factors (low precipitations and high temperatures), weathering of native rocks, and irrigation mismanagements, i.e., irrigation methods and water quality [14]. It has been also predicted that more than 50% of the arable land will be affected by salinity by 2050 [15]. The accelerated development of this threatening phenomenon currently represents a serious problem to the maintenance of the health and functionality of arid ecosystems and due to the significant impacts it has on land degradation and desertification, reduction of crop production, economic aspects [16], human wellbeing and sustainable development.

Soil salinity is a dynamic phenomenon presenting a serious challenge and requiring efficient quantification methods for mapping and monitoring in space and time [17,18]. Indeed, combating soil salinization should lead to enhanced agricultural productivity and profitability, and it should ensure food security. To remedy this situation in vulnerable landscape to salinity, there are methods available to slow down the processes and, sometimes, even reverse them [19]. However, remedial actions require reliable information to help set priorities and to choose the most appropriate intervention for a specific location. In affected areas, farmers, soil managers, scientists, and agricultural engineers need accurate and reliable information on the nature, extent, magnitude, severity, and spatial distribution of the salinity to take appropriate measures [20,21]. Unfortunately, soil salinity monitoring in space and time is complicated by the dynamic nature of salinity due to the influence of management practices and physical factors mentioned above. When the need for repeated measurements in time is multiplied by the extensive requirements of a single sampling period, the expenditures of time and effort with conventional soil sampling procedures increase proportionately [21]. At the international level, measuring electrical conductivity of extract from a saturated soil paste at the laboratory ($EC_{-Lab}$) is the most accurate method used for soil salinity assessment, as developed by the USDA [18,22]. However, the preparation of pastes and their extracts is a labor-intensive and expensive process, especially for regular monitoring over a long period and for comparisons over large areas [13,17,23]. Thus, in the field of soil salinity mapping, remote sensing offers significant scientific advantages over ground-based methods [18,24–26]. During the last three decades, remote sensing sciences, technology and image processing methods have outperformed these conventional methods [27–31]. They can bridge economic, scientific and practical considerations to extract accurate and relevant information. Moreover, their main advantages are represented by providing a unique opportunity for mapping large areas at relatively low cost and collecting and processing information at regular intervals that will support the monitoring process. These allow not only for the rapid appropriate remedial action to be taken but also for the monitoring of the effectiveness of any ongoing remediation or preventative measures, which facilitate management and decision-making policy [21,25,32–34].

Furthermore, in addition to the remote sensing sensor technology improvement and innovation, several image processing methods and models were developed and applied for soil salinity retrieval, such as spectral salinity indices [35–39], mixture-tuned matched filter approach [40], regression of multi-spectral bands [41–43], partial least square regression [37], multivariate adaptive regression splines [44], artificial neural network model [45], linear spectral mixture analysis [46], and machine learning regression [47,48], as well as several empirical, semi-empirical and physical predictive models [27–31,38,39,49,50]. These latest models were developed and tested to predict soil salinity in different semi-arid and arid geographic regions around the world, i.e., Latino-America (Mexico), the Middle East (Iraq), North and East Africa (Morocco and Ethiopia) and Asia (China). Likewise, these models integrate different spectral bands and are different to mathematical functions in terms of their conceptualization and modeling (stepwise, linear, polynomial second-order, logarithmic, and exponential functions). They were developed based on simulated spectral data and/or satellite images acquired with different sensors, such as Thematic Mapper (TM), Enhanced Thematic Mapper plus (ETM+), Operational Land Imager (OLI), Sentinel-2 Multispectral Instrument (MSI), Earth Observing One—Advanced Land Imager (EO-1-ALI) and WorldView-3. The present study focuses on validation and comparison among eight different empirical and semi-empirical models for soil salinity prediction and mapping in an arid landscape, exploiting two independent Landsat-OLI datasets: simulated and image data.

## 2. Materials and Methods

Figure 1 illustrates the methodologies used in the present study structured in four steps, exploiting two independent datasets: simulated and image data. For simulated data, a field survey was completed during over days, and 100 soil samples were collected with various degrees of soil salinity levels (i.e., extreme, very high, high, moderate and low) including non-saline. The bidirectional reflectance factor was measured above each soil sample in a goniometric laboratory using an analytical spectral device (ASD) FieldSpec-4 high-resolution (Hi-Res) spectroradiometer [51]. The required preprocessing steps to allow their meaningful and accurate use and comparison were then carried out. Indeed, all measured spectra were resampled and convolved in the solar-reflective spectral bands of an OLI sensor using the Canadian Modified Simulation of a Satellite Signal in the Solar Spectrum (CAM5S) [52] based on Herman radiative transfer code (RTC), assuring an accuracy of 0.6% in the visible and 1.1% in the infrared bands with the relative spectral response profiles characterizing the Landsat-OLI filters of each visible- and near-infrared (VNIR) and shortwave infrared (SWIR) bands. Meanwhile, the OLI image was acquired over the study site on 13th May 2017 at the same time as the field survey. This image was not cloudy or cirrus contaminated and was without shadow effects because significant topographic variations are absent in the study area. It was calibrated radiometrically and corrected atmospherically to transform the digital number to the ground surface reflectance. Finally, the standardized ground reflectance factors (simulated and image data) were converted in terms of the eight examined models. For validation and comparison, statistical fits were conducted using linear regression analysis ($p < 0.05$); therefore, coefficient of determination ($R^2$) and the root mean square error (RMSE) were calculated. It is important to note that subsequent to the spectroradiometric measurements, the soil chemical properties (cations and anions: $Ca^{2+}$, $Mg^{2+}$, $Na^+$, $K^+$, $Cl^-$ and $SO_4^{2-}$), the pH of saturated soil paste (pHs) and the electrical conductivity ($EC_{Lab}$) of the extract from saturated soil paste were measured in the laboratory, as well as the Sodium Adsorption Ratio (SAR) being calculated using the standard calculation procedure [23]. These parameters provided reliable information about the type and the degree of salinity and sodicity in each soil sample, and, thus, they help to understand the close relationship between the soil salt contents and their spectroradiometric behavior.

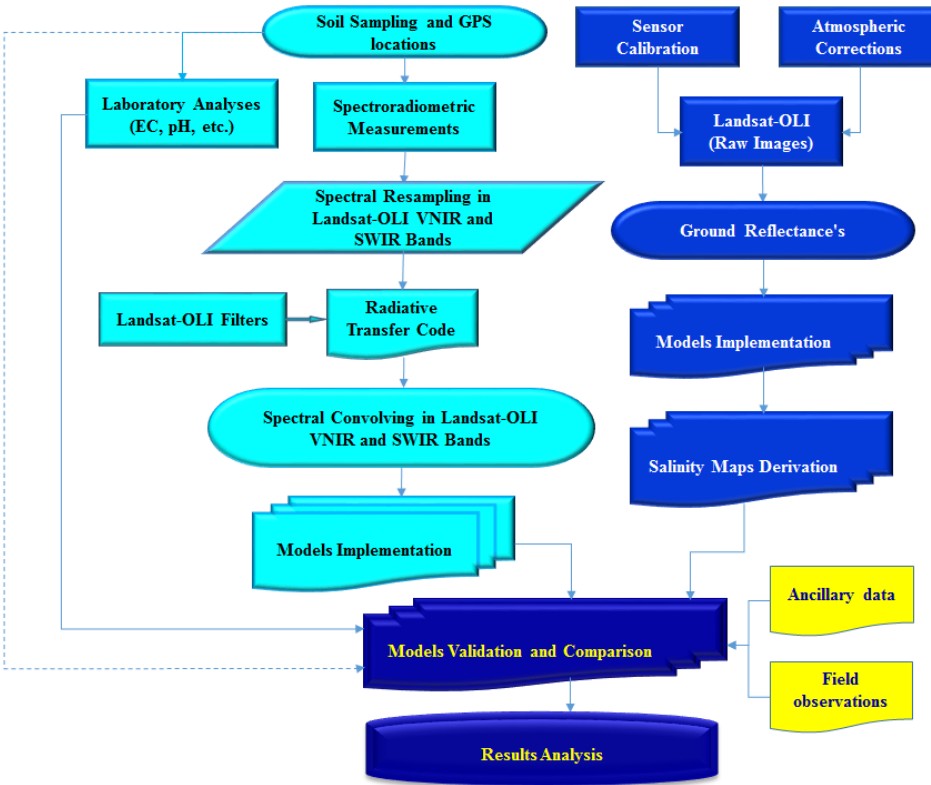

**Figure 1.** Methodology Flowchart.

*2.1. Study Site*

The state of Kuwait (Figure 2) situated in the north-western part of the Arabian Peninsula (28°45′ to 30°06′N, 46°33′ to 48°35′E) is characterized by an arid climate, very hot temperature during the summer season (46.8 °C) and irregular rainfall with an annual mean of 118 mm. The main geomorphological features characterized the study area are escarpments, sand dunes, sabkhas (pure salt accumulation), depressions, playas and alluvial fans [53,54]. These features are controlled by three types of surface deposits [55]. The first is represented by Aeolian deposits such as dunes and sand sheets. The second is defined by evaporites, such as halite (NaCl), gypsum ($CaSO_4.2H_2O$) and anhydrite ($CaSO_4$), as well as other salt deposits in the coastal and inland sabkhas. The third include fluvial deposits, such as pebbles and gravels, which are located along the Wadi channels. Each of these deposits has specific geomorphic characteristics based on their origin, topography, which is generally flat with a low relief, and climatic impacts [53,56]. Geologically, Kuwait stratigraphy consists of two stratigraphic groups: the Kuwait group and Hasa group [57] consisting of six formations, four of which are exposed in the outcrops represented by the Dammam, Ghar, Mutla and Jal-AzZor Formations. In the Hasa group, the Dammam Formation *(Eocene)* consists of white fine-grained cherty limestone and forms some karst. However, the three other formations are composed mainly of sandy limestone, calcareous sandstones, sand and clay. The soils of Kuwait are mostly categorized as sandy with limited organic matter, very low nutrient and very high amounts of calcareous materials. Moreover, a gatch layer occurs in many Kuwaiti soils, which is considered a calcic and/or gypsic pan [58].

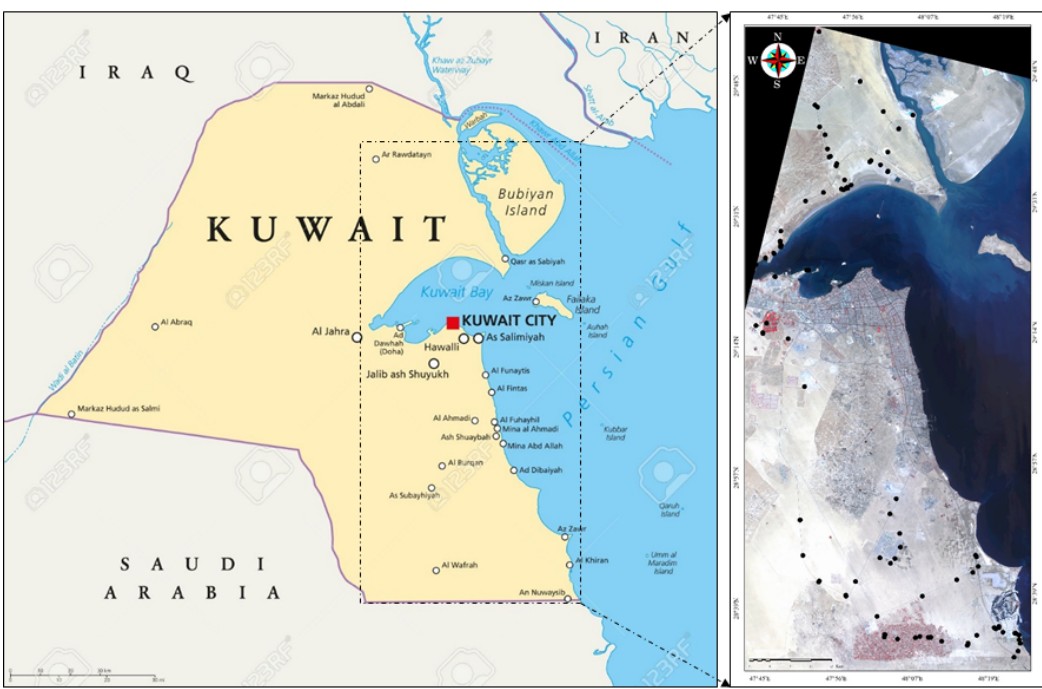

**Figure 2.** Study site and sampling point locations.

Based on the Landsat-OLI satellite orbit, more than half of the Kuwait territory (center and east) was covered as an experimental area for this study (Figure 2), including irrigated agricultural fields, desert land, urban areas, coastal zones, and low-land such as Bubiyan Island.

### 2.2. Soil Sampling and Laboratory Analysis

According to the United States Department of Agriculture [59], soils of Kuwait are mostly sandy with a very low organic matter and are infertile. They have been classified into two main soil orders: the Aridisols, occupying 70.8%, and the Entisols, occupying 23.2% of the area surveyed, while the other restricted and marginal groups represent the remaining percentage (~6%). These two soil orders are further classified into eight soil great groups based on morphology and mineralogical, chemical and physical characteristics [58,59]. The extreme soil salinity class (sabkhas) occurs in the Aquisalids soil great group on coastal flats and inland Playas, which contain very high salt contents and gypsum. The high soil salinity class is identified in Haplocalcid soils due to the layer of carbonate masses and salt contents. The moderate to low salinity class occurs in Petrocalcid soil, which is characterized by calcic hardpan overlying sandy to loamy soils and the presence of scattering halophytic plants.

Based on the fieldwork and soil map of Kuwait, the following soil salinity classes represented by the photos in Figure 3A–F were considered: non-saline, low, moderate, high, very high, and extreme salinity. A total of 100 soil samples representing these six classes were collected over four days (15th to 18th May 2017) during field visits with good spatial distribution over the study area (Figure 2). Samples were collected from the upper layer of the soil (0 to 5 cm deep considering an area of about 50 × 50 cm) and placed and numbered in plastic bags. In addition, each soil sample was physically described (color, brightness, texture, etc.), photographed and geographically localized using accurate GPS ($\sigma \leq \pm 30$ cm) connected in real-time to a GIS database integrating the OLI image (acquired two days before field survey, the 13th May 2017) for accurate spatial location and identification of each salinity class for the validation step.

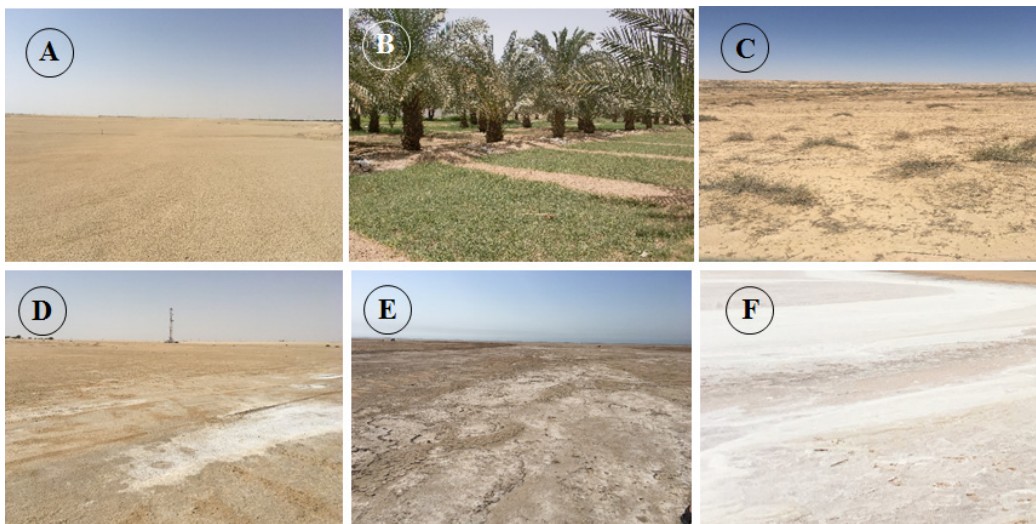

**Figure 3.** Photos illustrating the six considered soil salinity classes: non-saline (**A**), low (**B**), moderate (**C**), high (**D**), very high (**E**), and extreme salinity, which is sabkha (**F**).

In the laboratory, the considered soil samples were air-dried, ground, and passed through 2 mm sieve. The saturated soil paste extract method was utilized to measure the $EC_{-Lab}$ and pH. Moreover, the major soluble cations ($Ca^{2+}$, $Mg^{2+}$, $Na^+$, and $K^+$) and anions ($Cl^-$ and $HCO_3^-$) were measured in the soil paste extract, and the sodium adsorption ratio (SAR) was calculated. These analyses were carried out at the soil laboratory using methods that meet the current international standards in soil science [22,23,60]. The electrical conductivity of soil saturation extract (ECe i.e., $EC_{-Lab}$) is considered an accurate measure of soil salinity. The derived $EC_{-Lab}$ values for all sampling points were used for models validation using measured spectral reflectance data, as well as overlaid on the remote sensing predictive salinity maps ($EC_{-Predicted}$) derived from the OLI image in a GIS environment based on GPS locations for spatial correspondences between field sampling points and their homologous pixels for validation.

### 2.3. Spectroradiometric Measurements

Spectroradiometric measurements were acquired in a goniometric laboratory. The bidirectional reflectance spectra were measured above each air-dry soil sample using an ASD (Analytical Spectral Devices Inc., Longmont, CO, USA) FieldSpec-4 Hi-Res (high-resolution) spectroradiometer [51]. This instrument is equipped with two detectors operating in the VNIR and SWIR, between 350 and 2500 nm. It acquires a continuous spectrum with a 1.4 nm sampling interval from 350 to 1000 nm and a 2 nm interval from 1000 to 2500 nm. The ASD resamples the measurements in 1- m intervals, which allows the acquisition of 2151 contiguous bands per spectrum. The sensor is characterized by the programming capacity of the integration time, which allows an increase in the signal-to-noise ratio (SNR) as well as stability. The data were acquired at nadir with a field of view (FOV) of 25° and a solar zenith angle of approximately 5° by averaging forty measurements. The ASD was installed at a height of 60 cm approximately over the target, which makes it possible to observe a surface of approximately 700 cm$^2$. A laser beam was used to locate the center of the ASD-FOV over the center of each target. The reflectance factor of each soil sample (Figure 4) was calculated by rationing target radiance to the radiance obtained from a calibrated "Spectralon panel" in accordance with the method described by Jackson et al. [61]. Moreover, the corrections were applied for the wavelength dependence and non-Lambertian behavior of the panel [51,62,63].

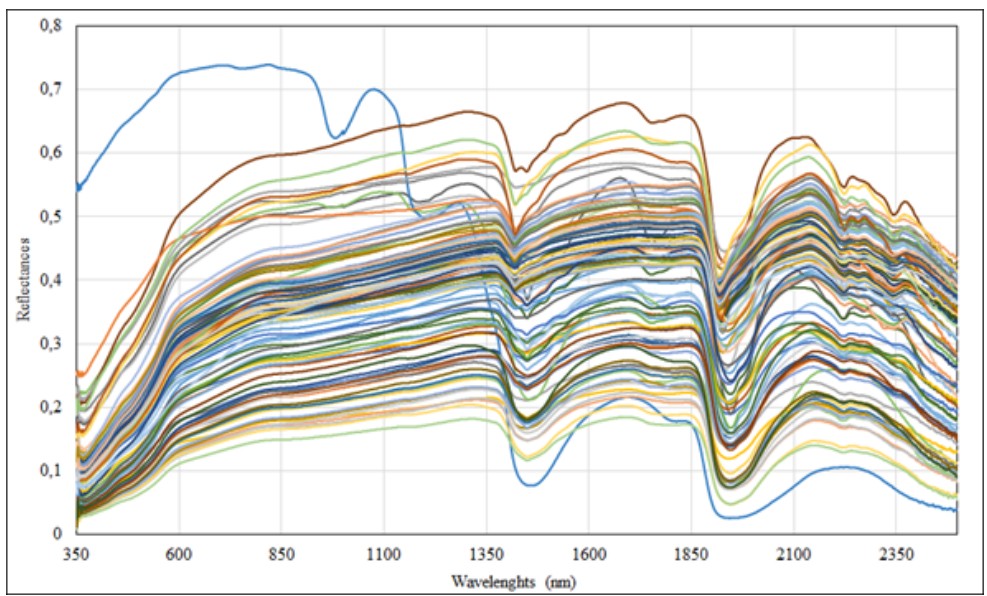

**Figure 4.** Spectral signatures of 100 soil samples with different degrees of salinity.

### 2.4. Landsat-OLI Simulated Data

The measured bidirectional reflectance factors using the ASD have a 1- m interval, which allows the acquisition of 2151 contiguous hyperspectral bands per spectrum. However, most multispectral remote sensing sensors measure the reflectance that is integrated over broad bands. Consequently, the measured spectra of each soil sample were resampled and convolved to match the OLI solar-reflective spectral response function characteristics in the VNIR and SWIR spectral bands, as shown in Figure 5. In this step, the resampling procedure considers the nominal width of each spectral band. Then, the convolution process was executed using the CAM5S RTC [52]. This fundamental step simulates the signal received by the OLI sensor at the top of the atmosphere from a surface reflecting solar and sky irradiance at sea level, considering the filter responsivities of the individual sensor band (Figure 5) and assuming ideal atmospheric conditions without scattering and absorption [64–67]. The reflectance values of 100 soil samples, with various salinity degrees, were simulated and generated at the sensor level in the OLI VNIR and SWIR spectral bands. Thus, the examined salinity models were calculated, and the predicted salinity classes (EC-$_{\text{Predicted}}$) were fitted with the salt contents measured in the laboratory (EC-$_{\text{Lab}}$) using linear regression analyses ($p < 0.05$). This statistical examination step was used to validate and to evaluate the strength of the models' capability to discriminate among various salinity classes.

### 2.5. Landsat-OLI image

The Landsat scientific collaboration program between NASA and USGS has conducted the continuous recording of the Earth's surface reflectivity from space over nearly five past decades. It supports global moderate resolution data collection, distribution and archive of the Earth's continental surfaces [68] for research, environmental application and climate change impact analyses at the global, regional and local scales [69–71]. On 11th February 2013, the polar-orbiting Landsat-8 satellite was launched, transporting two push-broom instruments: OLI and Thermal Infrared Sensor (TIRS). The OLI sensor collects land surface reflectivity in the VNIR, SWIR and panchromatic wavelengths with a FOV of 15° covering a swath of 185 km with 16 days' time repetition at the equator. The band passes are narrower in order to minimize atmospheric absorption features [72], especially the NIR spectral band (0.865 µm). Two new spectral bands have been added: a deep blue visible shorter wavelength (band 1: 0.433–0.453 µm), designed specifically for water resources

and coastal zone investigation, and a new SWIR band (9: 1.360–1.390 μm) for the detection of cirrus clouds. Moreover, the OLI design results in a more sensitive instrument with a significant amelioration of the SNR radiometric performance quantized over a 12-bit dynamic range (level 1 data); raw data are delivered in 16 bit. This SNR performance and improved radiometric resolution provide a superior dynamic range, reduce saturation problems, maximize the range of land-surface spectral radiance and, consequently, enable better characterization of land cover conditions [73]. According to Gascon et al. [74] and Markham et al. [75], in terms of orbit reflective wavelength calibration, Landsat-OLI is better by 3%. In the present study, the used OLI image was acquired approximately at the same time as the field soil survey, which was carried out on the 13th May 2017.

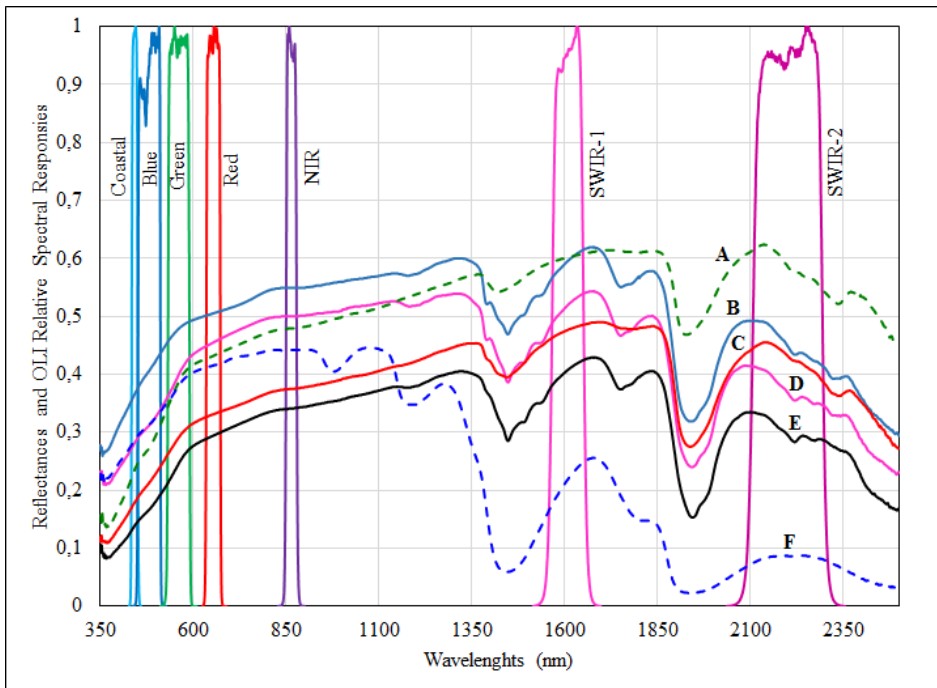

**Figure 5.** Landsat-Operational Land (OLI)-relative spectral response profiles characterizing the filters of each visible- and near-infrared (VNIR) and shortwave infrared (SWIR) spectral band, and the spectral signatures of six soil samples with different electrical conductivity (EC-$_{Lab}$) values: non-saline (**A**), low (**B**), moderate (**C**), high (**D**), very high (**E**), and extreme salinity (**F**).

### 2.6. Image Data Pre-Processing

Before the processing operation and accurate information extraction, pre-processing steps are required [64]. They are related to sensor drift radiometric calibration and atmospheric corrections (scattering and absorption) [76,77]. The OLI raw scene measured at the top of the atmosphere (TOA) was transformed into apparent equivalent radiance using appropriate radiometric absolute calibration parameters (gain and offset) delivered by the EROS-USGS center for each spectral band. Then, the extra-atmospheric irradiance, the solar zenith angle and the Earth–sun distance (in astronomical units) were used to transform the apparent radiance to the apparent reflectance. Moreover, the CAM5S model [52] was used for atmospheric conditions simulations to calculate all the requested atmospheric correction parameters in the OLI spectral bands. This CAM5S model simulates the signal measured at the TOA from the Earth's surface reflecting solar and sky irradiance at sea level while considering sensor characteristics, such as the band passes of the solar-reflective bands, satellite altitude, measured atmospheric conditions during the satellite overpass, atmospheric model, sun and sensor geometries, and terrain elevation. Consequently, all of the requested atmospheric correction parameters were calculated and applied to transform the apparent reflectance at the TOA to ground reflectance. Sensor calibration and atmospheric

interferences were then combined and corrected in one step using PCI-Geomatica [78] to preserve the radiometric integrity of the image data [79].

### 2.7. Soil Salinity Models and Image Processing

Soil salinity is modeled for global, regional and local scales around the world, considering several environment and using, respectively, coarse, moderate and high spatial resolution, i.e., MODIS, Landsat, and Worldview [53,80]. Consequently, numerous empirical, semi-empirical and physical models were developed for soil salinity mapping and monitoring. The present study is the first attempt to validate and compare eight models developed for salt-affected soil mapping in semi-arid and arid regions, i.e., Latino-America, the Middle-East, Africa and Asia [9,20,24,49,50,81–84]. In addition to the geographic location, these models integrate different spectral bands and are different to mathematical functions in terms of their conceptualization, such as stepwise, linear, second-order polynomial, logarithmic, and exponential functions.

For a Mexican environment, the first developed model is based on the exponential function [49] integrating in its equation the EC-$_{Lab}$, the SAR, the spectral responses of the bare soil and vegetation cover fraction calculated from the Normalized Difference Vegetation Index (NDVI) and the Combined Spectral Response Index (COSRI). The second model was developed to map spatiotemporal soil salinity variation of irrigated agricultural land in Morocco based on the first linear regression and integration of the four visible spectral bands of Landsat-OLI [50]. For Ethiopian agricultural land, the third proposed model is also based on linear regression by integrating the NIR and red spectral bands [83]. The fourth model was developed to map the largest "Sda" saline–alkaline regions in China [24] by exploring only the blue band and the stepwise linear regression. The fifth, sixth and seventh models were developed for arid land in Iraq [84], considering only the red and NIR bands but with certain particular difference conditions in terms of experimentation for specific scales and applications. Indeed, the fifth model was developed for regional scale based on a logarithmic function, integrating the soil apparent salinity (EM$_v$) measured in vertical direction and the vegetation cover information based on the Generalized Difference Vegetation Index (GDVI). The sixth model is a simple linear logarithmic model based only on the GDVI and does not consider EM$_v$ variable. In contrast, developed specifically for the local scale and considering vegetated areas, the seventh model relies on both logarithmic and exponential functions integrating the GDVI and the apparent soil salinity (EM$_H$) measured in the horizontal direction. Finally, in the semi-arid environment in Morocco, the eighth model was developed for slight and moderate salinity prediction over irrigated agricultural land based on the SWIR spectral bands, EC-$_{Lab}$ and a second-order polynomial function [20]. Tested in the Middle-East, this model discriminates significantly among several soil salinity classes and highlights the remarkable sabkha areas, i.e., extreme salinity [32,81]. All of these considered models were implemented and calculated based on a simulated reflectance factor and the OLI image using EASI-modeling of PCI-Geomatica image processing software [78].

$$\text{Model} - 1 = 348.104 * \text{Exp}^{-18.372 * \text{COSRI}}$$
$$\text{COSRI} = \left( \frac{\text{B} + \text{G}}{\text{R} + \text{NIR}} \right) * \text{NDVI} \tag{1}$$
$$\text{NDVI} = (\text{NIR} - \text{R}) / (\text{NIR} + \text{R})$$

$$\text{Model} - 2 = 13.56 * OLI_{-SI} - 2.24$$
$$OLI_{-SI} = \left( CB^2 * 50 \right) - (B + G + R) \tag{2}$$

$$\text{Model} - 3 = -0.706 + 7.519 * SI$$
$$SI = \sqrt{R * NIR} \tag{3}$$

$$\text{Model} - 4 = -3.35 + 0.56 * B \tag{4}$$

$$\text{Model} - 5 = 0.0005 * \text{EM}_V^2 - 0.0779 * \text{EM}_V + 12.655$$
$$\text{EM}_V = 66.338 - 258.114 * \ln(\text{GDVI}) \tag{5}$$
$$\text{GDVI} = \left(\text{NIR}^2 - \text{R}^2\right) / \left(\text{NIR}^2 + \text{R}^2\right)$$

$$\text{Model} - 6 = -2.87 - 23.27 * \ln(GDVI) \tag{6}$$

$$\text{Model} - 7 = 0.0002 * EM_H^2 + 0.0956 * EM_H + 0.0688$$
$$\text{EM}_H = -606.197 - 460.03 * \ln(\text{GDVI}) + 245.086 * \text{Exp}(\text{GDVI}) \tag{7}$$

$$\text{Model} - 8 = \text{C}^{\text{st}} * \left[4521 * (\text{SSSI2})^2 + 124.50 * (\text{SSSI2}) + 0.41\right]$$
$$\text{SSSI2} = \frac{(\text{SWIR1}*\text{SWIR2}) - (\text{SWIR2}*\text{SWIR2})}{\text{SWIR1}} \tag{8}$$

where C, B, G, R and NIR are the ground reflectance in the coastal, blue, green, red and near-infrared spectral bands, respectively. The SWIR1 and SWIR2 are the ground reflectance in shortwave infrared spectral bands, i.e., OLI-6 and OLI-7 bands, respectively. SI means a salinity index. Cst is a scaling factor between the ground based-measurements and the use of satellite image data (must be calculated for each application case) [85].

### 2.8. Statistical Analysis

Remote sensing for soil salinity mapping is intrinsically associated with uncertainties; therefore, a well-designed validation method is required. In this study, statistical analyses were computed using "Statistica" software. Various statistics were calculated between the electrical conductivity of ground sampling points obtained from the laboratory analysis (EC-$_{\text{Lab}}$) and the predicted values (EC-$_{\text{Predicted}}$) derived from simulated and OLI image data to assess the performance of each examined model. For this validation step, EC-$_{\text{Lab}}$ and EC-$_{\text{Predicted}}$ were compared using the 1:1 line. Ideally, observed and predicted values should have a correspondence of 1:1. Based on a linear regression, the $R^2$ was calculated to evaluate the strength of the linear relationship between the two considered variables values (EC-$_{\text{Lab}}$ and EC-$_{\text{Predicted}}$). Indeed, $R^2$ provides a measure of the degree to which a model's predictions are error free. It ranges between 0 and 1, with 1 indicating a perfect match between observed and predicted values. Systematic linear over-predictions or under-predictions generate characteristic variations in the slope and intercept values, which can help to interpret the major sources of error and the potential of the considered models for salinity prediction and mapping. Moreover, the RMSE was used as an overall error to supplement the coefficient of determination described above. It is an indicator for model errors and performance, estimating the absolute error between EC-$_{\text{Lab}}$ and EC-$_{\text{Predicted}}$ values. This residual error also quantifies the 1:1 relationship between observed and predicted values. These variables were calculated as follows [86]:

$$R^2 = 1 - \left[\frac{\sum\limits_{i=1}^{n}(P_i - O_i)^2}{\sum\limits_{i=1}^{n}\left(|P_i'| + |O_i'|\right)^2}\right] \tag{9}$$

$$RMSE = \sqrt{\frac{\sum_{i=1}^{n}(P_i - O_i)^2}{n}}. \tag{10}$$

where $P_i$ is the predicted value at sample $i$, $O_i$ is the observed value at sample $i$, $P_i'$ is the difference between $P_i$ and the average of the predicted values, $O_i'$ is the difference between $O_i$ and the average of the observed values, and $n$ is the number of values.

### 3. Results and Discussion

#### 3.1. Spectra and Soil Laboratory Analyses

Figure 4 shows that, overall, the spectral signatures of the considered 100 soil samples are controlled by the type of salt existing in each sample, such as sulphates; chlorides;

and/or carbonates of mainly sodium, calcium and magnesium. These spectra showed different amplitudes and several absorption features depending on the chemical compositions and mineral assemblages of the existing salts in the selected soil samples. Moreover, the spectral signatures are also influenced by several factors, such as mineralogical composition, impurity, structure, and texture of the soil and salt crystals, and the soil optical properties (color brightness, and roughness), particularly in the VNIR spectral domain [25]. Furthermore, the laboratory analyses of all soil samples revealed that the moisture content values are distributed in a very limited range between 0 and 0.05%, thus minimizing the impact of moisture content on the measured spectra (Figure 4). In fact, only weak absorption bands near 970, 1160, 1350, 1800, and 2208 nm were observed in some samples (atmospheric water vapor absorption features at 1440 and 1920 nm are not considered in this analysis). In contrast, the other absorption features are automatically linked to the salt mineralogy, particularly gypsum, sodium chloride (halite), calcium carbonate and sodium bicarbonate, which reveals significant absorption features at 980, 1000, 1190, 1210, 1400, 1450, 1490, 1540, 1748, 1780, 1800, 1900, 1945, 1975, 2175, 2215, 2265 and 2496 nm [25]. These observations corroborate the findings of other studies [25,45,87].

EC$_{-lab}$ was measured to quantify the salt content, to classify the soil salinity classes and to validate the examined models. Moreover, soil texture analysis was accomplished for physical characterization of soil samples. Table 1 summarizes the six classes of salinity according to the EC$_{-Lab}$ values and presents the mean values of cations and anions in each class, and Figure 5A–F illustrate their spectral signatures. According to the results of the laboratory analyses, it can be observed that the EC$_{-Lab}$ values are distributed progressively in a wider range between 1.6 and 700 dS.m$^{-1}$ for the 100 considered soil samples. The lowest values represent the samples of agricultural fields, while the highest indicate the sabkha "salt scald" consisting of pure salt. These soil samples present higher quantities of chloride (Cl$^-$: 9.6 to 3932 meq/L), sodium (Na$^+$: 23 to 3615 meq/L), magnesium (Mg$^{2+}$: 7.8 to 1118 meq/L) and calcium (Ca$^{2+}$: 39 to 230.4 meq/L) than other elements. Indeed, Figure 6 illustrates that the dominant ions in the soil samples are chloride (Cl$^-$) and sodium (Na$^+$) showing, respectively, an R$^2$ of 0.98 and 0.87 with EC$_{-lab}$. Meanwhile, a weak relationship with Ca$^{2+}$ (R$^2$ of 0.23) and a moderate relationship with Mg$^{2+}$ (R$^2$ of 0.48) and K$^+$ (R$^2$ of 0.46) can be observed. The main sources of Cl$^-$ in the soil are from seawater (level rise and spray), precipitation, salt dust, irrigation and fertilization. In contrast, parent material, pedogenic processes, irrigation with saline–sodic waters and inappropriate soil drainage are the main sources of Na$^+$. Likewise, it was observed that the EC$_{-Lab}$ and SAR increased gradually and very largely from non-salinity (EC$_{-Lab}$: 1.6 dS.m$^{-1}$, SAR: 0.4) to extreme salinity in sabkha (EC$_{-Lab}$: 700 dS.m$^{-1}$, SAR: 445), yielding an R$^2$ of 0.70 between them. Notably, high values of SAR are attributed to the presence of a high concentration of Na$^+$ in the soil relative to Ca$^{2+}$ and Mg$^{2+}$.

**Table 1.** Laboratory analysis of the six soil classes that are presented in Figures 3 and 5.

| Sample | Salinity Classes | EC$_{-Lab}$ dS.m$^{-1}$ | pHs | Ca$^{2+}$ | K$^+$ | Mg$^{2+}$ | Na$^+$ | Cl$^-$ | HCO$_3^-$ | CaCO$_3$(%) | SAR (mmoles/l)$^{0.5}$ |
|--------|------------------|------|-----|------|-----|------|------|------|------|----------|----------|
| | | | | | | meq/L | | | | | |
| A | Non-Saline | 2.4 | 7.7 | 18.9 | 0.6 | 3.6 | 5.5 | 8 | 4 | 26.5 | 1.6 |
| B | Low | 6.7 | 7.7 | 67 | 2.3 | 12 | 23 | 38 | 9.1 | 19 | 3.7 |
| C | Moderate | 11.8 | 7.7 | 45 | 7 | 14 | 49 | 70 | 10 | 26 | 9.1 |
| D | High | 38.4 | 7.3 | 146 | 310 | 100 | 258 | 350 | 6 | 12.5 | 23.3 |
| E | Very High | 48.8 | 7.4 | 78 | 15 | 19 | 325 | 590 | 4 | 19.5 | 46.8 |
| F | Extreme | 400.3 | 7.0 | 230.5 | 97 | 1118 | 3615 | 3932 | 6.6 | 22 | 139.2 |

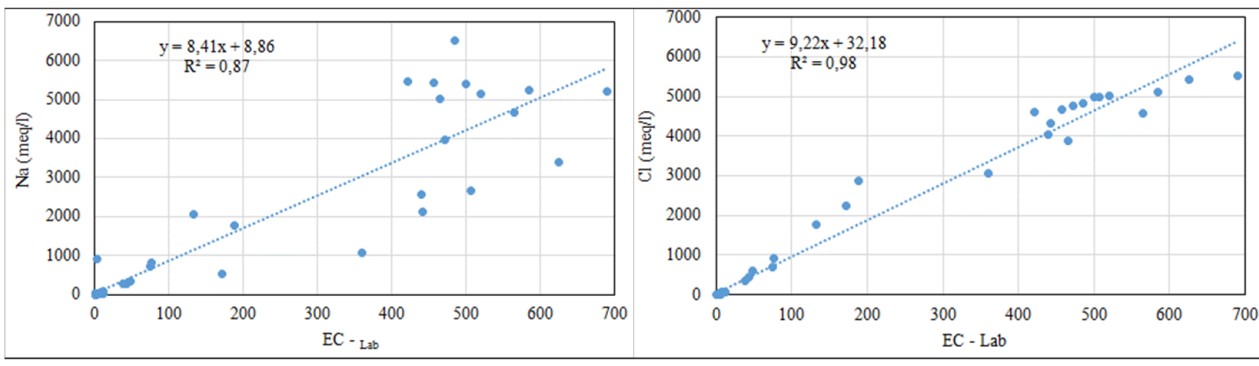

**Figure 6.** Linear regression between EC-$_{Lab}$ and chemical parameters (Na$^+$ and Cl$^-$).

Sequentially, the soil pH values ranging from 7 to 7.7 indicated a slight alkaline reaction due to the presence of bicarbonate (HCO$_3$$^-$) in the soils with a range of from 4 to 10 meq.l$^{-1}$. The results also showed that CaCO$_3$ ranged from 12.5 to 26 %, showing calcareous soil and parent materials, which significantly occurs in the arid regions due to the scarcity of rainfall. The calcareousness could be partially due to water quality through an extended period of irrigation, where CaCO$_3$ precipitated due to combined bicarbonates and carbonates with Ca, especially in dry conditions. The results of laboratory analyses showed the dearth (<2.6%) of organic matter (OM) in all soil samples, with an average of 0.58 %. This corresponds to the total organic carbon percentage of 1.6 %, with an average of 0.34 %. Soil texture analysis showed an increase in salt contents with a decrease in soil particle size.

### 3.2. Model Validation and Comparison Based on Simulated Data

Figure 7 illustrates the relationship between EC-$_{Lab}$ and EC-$_{Predicted}$ derived from the eight examined models based on simulated data. Except model 8, the other seven models predict the salt content in the soil differently and are far from the observed truth in the field. For instance, model 1 discriminates the six existing salinity classes in two main classes only, and it reverses the logic sense of prediction with severe confusion between classes (Figure 7a). The first predicted class largely overestimates the salt content in the soil, and it combines the samples of non-saline, low, moderate and high salinity classes. For instance, soil samples with the EC-$_{Lab}$ range varying between 1.6 and 200 dS.m$^{-1}$ are predicted between 70 and 240 dS.m$^{-1}$. In contrast, the second class group samples with very high and extreme salinity (EC-Lab $\geq$ 200 dS.m$^{-1}$) are unfortunately severely underestimated in a very limited range solely between 100 and 150 dS.m$^{-1}$. Moreover, as depicted by Figure 7a, the scatter plot resulting from this model shows a very poor linear fit (1:1 line) between EC-$_{Lab}$ and EC-$_{Predicted}$, yielding an R$^2$ of 0.33 and the highest RMSE of 275%. Accordingly, this model is very ineffective for predicting salt content in the soil in an arid environment. Likewise, model 2 also proved to be a poor predictor, because its scatter plot shows serious confusion within the predicted soil salinity classes (Figure 7b). Overall, the cloud of sampling point shows that no statistical relationship exists between the observed and predicted variables. Indeed, samples with high, very high and extreme salt content values (100 dS.m$^{-1}$ $\leq$ EC-$_{Lab}$ $\leq$ 700 dS.m$^{-1}$) are predicted as representing a single and non-saline class with values ranging between 1.6 and 5 dS.m$^{-1}$. Contrarily, the non-saline, low and moderate salinity samples (1.6 $\leq$ EC-$_{Lab}$ $\leq$ 15 dS.m$^{-1}$) are estimated as belonging to a high salinity class (15 $\leq$ EC-$_{Predicted}$ $\leq$ 55 dS.m$^{-1}$). The slope ($-0.02$) and the intercept (8.62) corroborate these observations by expressing a very large deviation from the 1:1 line (Figure 7b). Consequently, the statistical fit revealed a non-significant R$^2$ (0.26) concurrent with very strong RMSE (260%).

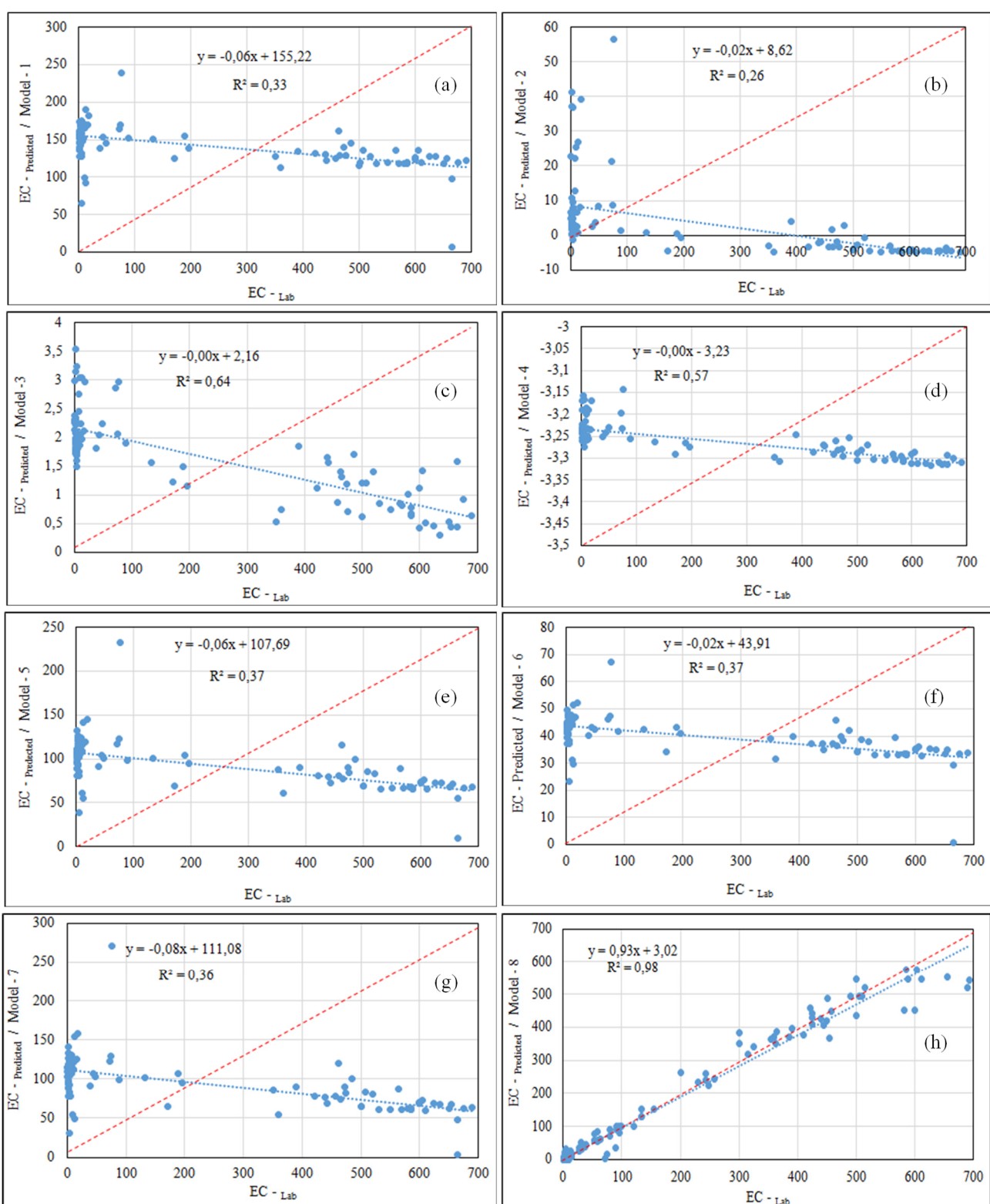

**Figure 7.** Relationship between EC-$_{Lab}$ and predicted salinity (EC-$_{Predicted}$) derived from models based on simulated data.

Except the non-saline and low salinity classes, regardless of the salt content observed in the considered soil samples (1.6 ≤ EC-$_{Lab}$ ≤ 700 dS.m$^{-1}$), the predictions of model 3 are not realistic and do not reflect the field truth. The predicted values oscillate between 1 and 2 dS.m$^{-1}$ for moderate, high, very high and extreme salinity classes, and between 1.5 and 3.5 dS.m$^{-1}$ for non-saline and low salinity (Figure 7c). However, although the

statistical fit between the observed and predicted values reveals an acceptable coefficient of determination ($R^2$ of 0.63), the calculated RMSE remain very high (253%). Figure 7d illustrates the scatter plot generated by model 4, showing a similar trend to that of model 3. It predicts the salt content negatively with a very narrow and limited range of values between $-3.3$ and $-3.15$ dS.m$^{-1}$, respectively, for non-saline and extreme salinity (sabkha). Compared to the ground reference observations, the statistical fit of model 4 generated an $R^2$ of 0.57 and a very strong RMSE (253%), confirming that it is a poor predictor of soil salinity in arid environments. Furthermore, Figure 7e,g illustrate the scatter plots generated with models 5, 6 and 7. They show the same trend depicting a poor relationship between observed and predicted values with insignificant $R^2$ (0.37) and a very high RMSE (270%). The slope and the intercept of these scatter plots corroborate these results by expressing a strong deviation from the 1:1 line. These three models, models 5, 6 and 7, severely underestimate the high, very high and extreme salinity classes, while they largely overestimate the non-saline, low and moderate salinity. The results obtained in this section—based on simulated data of the OLI sensor and considering only bare soils devoid of vegetation cover—reveal that these models are not quite suited for salt-affected soil mapping in an arid landscape.

Finally, the scatter plot established based on model 8 (Figure 7h) illustrates a good fit to the 1:1 line, particularly between 0 and 500 dS.m$^{-1}$, with a slope equal to 0.93 and an intercept of 3.02. It is also observed that the extreme salinity samples (EC-$_{Lab} \geq 500$ dS.m$^{-1}$) are slightly underestimated due to some saturation for this extreme class, resulting in the data not fitting the 1:1 line very well. This could be due to the presence of low solubility sulphate (gypsum) and carbonate/bicarbonates (nahcolite, natron and trona) minerals in conjunction with a high-solubility (NaCl-halite) mineral precipitated at the surface, hence diluting the salinity prediction. However, despite this minor variation, this validation procedure of comparison with EC-$_{Lab}$ shows that model 8 is generally a good predictor of soil salinity in an arid landscape, achieving an $R^2$ of 0.98 and an RMSE ranging between 5% and 10% for non-saline and extreme salinity classes, respectively. According to the results of these simulations, only model 8 provides satisfactory results for soil salinity prediction in an arid landscape.

### 3.3. Models Validation and Comparison Based on Visual Interpretation

To understand which model is relevant to the reliable and accurate predication of salt-affected soil classes in the arid landscape based on the Landsat-OLI image, validation and comparison procedures were undertaken visually by comparison with observations from the field visit and ancillary data (soil, geology, geomorphology and water table maps) and statistically based on the EC-$_{Lab}$ derived from laboratory analyses. Figure 8 illustrates the obtained soil salinity maps from the eight examined models, which are displayed in pseudo-color. Their histograms were threshold based on the defined six major salinity classes, including non-saline (in blue), low (in cyan or sky-blue-green), moderate (in clear green), high (in yellow), very high (in orange-red), and extreme salinity (red-purple).

The established map based on model 1 does not reflect the field truth—it discriminates only among four salinity classes instead of six (Figure 8a). The non-saline class, indicated in in blue, is located in desert areas in the south and the south-west of the study site (border with Saudi-Arabia). Additionally, the agricultural lands in the north (Abdali region) and south (Al-Wafra region) are classified as non-saline. The low salinity class, indicated in cyan, is located in areas along the coastal zones of Kuwait Bay and in lowlands (east and north of the study site), while Bubiyan Island displaying the moderate salinity class in green is a wrong prediction according to the field observations. Indeed, this Island area is a sabkha characterized by very low slopes (flat land), and the water table is very near the surface (remaining within 1.00 m from surface), promoting the accumulation of extreme contents of soluble salts and salt crust at the surface.

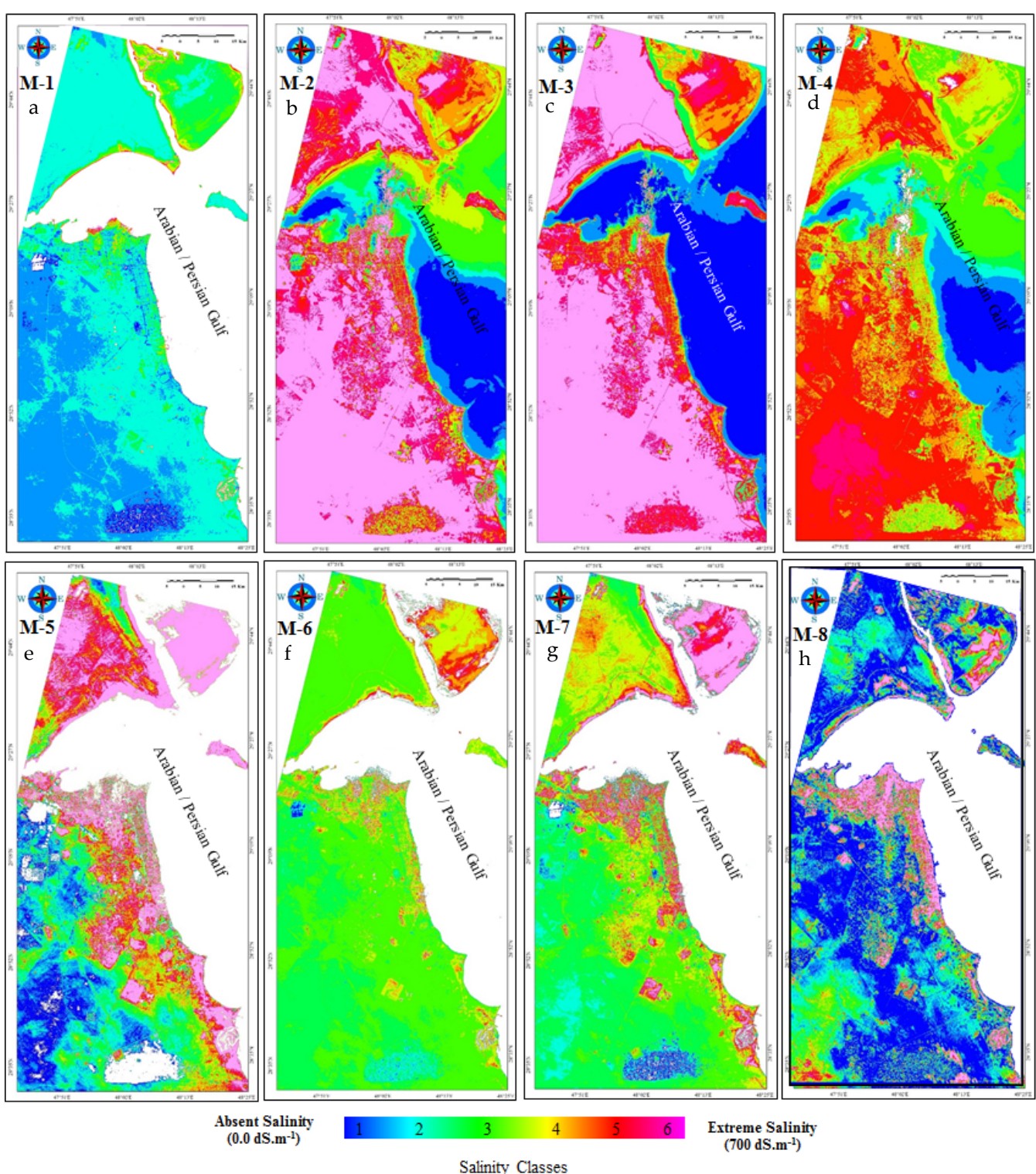

**Figure 8.** Derived soil salinity maps applying eight models.

Models 2 and 4 integrating the blue and the four visible bands, respectively, produced maps describing the soil salinity classes with major weaknesses and very considerable errors. In fact, only two classes (very high and extreme) are discriminated instead of six accounts for the entire study area (Figure 8b,d). In addition, agricultural fields that are characterized by low or non-saline classes were predicted as belonging to the moderate class by model 4 and belonging to the very high saline class by model 2. Moreover, the shallow seawaters in Kuwait bay were depicted wrongly as moderate and high soil

salinity classes by the both models (2 and 4) due to the blue band, which is sensitive to water mapping. In contrast, the derived map applying model 3 (Figure 8c) predicted the soil salinity classes almost similarly to model 2, but it described the aquatic surfaces as belonging to the non-saline class.

Furthermore, unlike the maps established by models 2 and 4 characterizing the desert areas (south and west of study site) as belonging to the extreme salinity class, that obtained by model 5 described these areas as belonging to numerous classes, namely, non-saline, low and moderate salinity (Figure 8e). In contrast, along the coastal zones of Kuwait Bay, the lowlands in the north and the Bubiyan Island were classed as possessing very high and extreme salinity. Although these two classes are mapped somewhat close to reality, unfortunately, some imperfections are noted because their spatial distribution is not consistent with the field truth. Indeed, an exaggerated overestimation was observed for these two classes. Moreover, the produced maps by models 6 and 7 (Figure 8f,g) represent the spatial distribution of salinity in a relatively similar way, with a high and dominant representativeness of the moderate class that covers most of the study area. However, while model 7 highlights the presence of the extreme salinity class (along the coastal areas, in the lowlands and on Bubiyan Island), its ability to reflect the field truth remains very limited.

Differently, visual analysis of the map derived by model 8 reveals perfect conformity with the ground truth. This map can reliably predict the six existing soil salinity classes in the study area. For instance, the extreme salinity class, depicted in red-purple in Figure 8h, is a cemented coastal deposit of pure salt and marine sand forming a sabkha, which occurs in the Haplosalids soils. This class is located in the coastal flat zones of Kuwait Bay, in lowlands, on Bubiyan Island and in playas above the high tide level; it is also characterized by very low topography (<1.2 m) where the water table is very near to the surface, remaining within 1.00 m from surface. These sabkha zones create a chemically aggressive environment and lead to a soil condition with high contents of soluble salts. These are established on natural lands (loamy and sandy, highly gypsiferous) and are devoid of any vegetation. According to the laboratory analyses, the salt content in this sabkha class is dominated by sulfates; chloride; and carbonates of sodium, calcium and magnesium. Seawater intrusion and subsequent evaporation results in surface salt crusts, especially in these zones of capillary rise where the slope is near zero, thus facilitating water catchment and retention. This situation generally occurs in a dry climate, such as in the case of Middle Eastern countries, where the total annual precipitation is very low (~100 mm/year) and the temperature is very high most of the year, thereby accelerating the evapotranspiration and the salt accumulation at the soil surface.

The very high and high salinity classes, depicted in yellow and orange-red, in Figure 8h, are located also in the coastal zones with relatively low slopes in Bubiyan Island, as well as in some inland areas. These classes are linked to Aquisalids soils, which are sandy to clayey with a layer of gypsum crystals and are poorly drained with a layer of salt accumulation near the surface. In addition, the water table remains within the upper 1.0 m of the surface. According to field visit and observations, the very high salinity class presents other crustal features, often encrusted with an efflorescence of salt crystals and a well-developed platy structure, which looks like the creation of a new sabkha. The high salinity class is composed of fine, white, sand-sized shell gravel and gravelly sand in which the surface layers are sometimes cemented by salt. According to laboratory texture analyses, this area is a mixture of medium and fine sand-grade quartz with moderate amounts of carbonate. The soil surface becomes cemented, forming a strong white-brown crust of salt (~5 cm). In other places, the immediate top surface becomes indurated into a brittle crust, which is ridged into many undulations with an amplitude of 3 to 5 cm. Moreover, the areas of high and very high salinity classes are also completely devoid of vegetation.

The moderate salinity class—depicted in green in Figure 8h and located in the northwest and southwest (border with Saudi Arabia)—is associated with sites of oil industry infrastructure and field operations. In these locations, the water table is at a depth greater

than 80 m, and the terrain elevations are higher than 150 m. This class is also located in the highest areas of Bubiyan Island, and it is composed of two group of soils Petrocalcids (sandy to loamy soil overlying a calcic hardpan) and Haplocalcids (sandy to loamy soils with a layer of calcium carbonate crystals). These soils are characterized by very poor organic matter content and very sparse and scattered clumps of halophytes. As discussed previously, according to the field visit, laboratory analyses and ancillary data (soil, geology, geomorphology and water table maps), we found that these four saline classes (extreme, very high, high and moderate) are often rich in sodium chloride and calcium sulfate and are associated with shales and marls, limestone and dolomitic limestone. Their gypso-saline characteristics cause the formation of salt strata that are thrusted upward to the surface from the underlying salt bed. These saline formations are often associated with anhydrite, gypsum, sulfur and paleo-lagoonary sedimentary rocks.

The low salinity class is located in the agricultural lands in the north (Abdali region) and south (Al-Wafra region) of the study area due to brackish water used for irrigation. This class includes only the cultivated areas in Kuwait (1% of the total area of the country), which are totally equipped with drip irrigation systems and are mainly used for the cultivation of date palms, alfalfa, vegetables and fodder crops (photo B in Figure 3). This class is dominated by two groups of soils (Torripsamments and Haplocalcids) located in flat areas with a low altitude (~20 m). The Torripsamments is a native soil to Kuwait that is generally non-saline with low fertility potential, and it is classified as a sandy or loamy sand. The Haplocalcids is a sandy to loamy soil that has a layer of carbonate masses. Finally, the non-saline class is composed of two main soil groups (Torripsamments and Petrogypsids) i.e., sandy to loamy soils overlying a petrogypsic horizon with relatively high topography ($\geq$50 m) and a moderately deeper water table (>30 m). Furthermore, the water surfaces of the Arabian Gulf (Kuwait Bay) are predicted as belonging to the non-saline class, which is true, since this model must be sensitive only to the soil salinity for which it was developed.

### 3.4. Model Validation and Comparison Based on Image Data

For validation and accuracy assessment of these eight derived salinity maps (Figure 8), the geographic locations of the 100 soil samples were superimposed on each map using the GPS coordinates. Afterward, homologous points between the field and maps were selected, and statistical fits (significance level of $p < 0.05$) were computed similarly to the simulated data between EC-$_{Lab}$ and EC-$_{Predicted}$. The only difference is that for the simulations, the EC-$_{Predicted}$ was implemented from the spectra of soil samples measured under an instantaneous field of view (IFOV) of 700 cm$^2$ in a controlled environment and observing solely bare soil. In contrast, for the OLI image, the EC-$_{Predicted}$ was calculated from pixels through an area of 900 m$^2$ with a mixture of soils from different horizons (texture, organic matter, color, brightness, etc.), and probably with the presence of very scattered vegetation cover (senescent and/or green). Figure 9 illustrates the relationship between EC-$_{Lab}$ and EC-$_{Predicted}$ derived from the OLI image when applying the considered eight models. The first major observation is that, except model 8, the models do not follow the regular and logic linear fit 1:1 line. A serious and strong overestimation can be observed for the non-saline, low and moderate salinity classes, while very significant saturation is found for the high, very high and extreme salinity classes.

Model 1 and model 5 are based on exponential and logarithmic functions, respectively. They integrate both the bar soil spectral responses and the vegetation cover fraction derived from the NDVI and GDVI based on VNIR bands. Such spectral domains make the two models very sensitive to the soil color and brightness [88]. Hence, this situation introduces very severe confusion and ambiguity between the predicted soil salinity classes. For instance, model 1 does not allow accurate discrimination among the six soil salinity classes (Figure 9a), and it predicts the non-saline and low (EC-$_{Lab} \leq 8$ dS.m$^{-1}$) as a moderate and very high salinity classes ($10 \leq$ EC-$_{Predicted} \leq 70$ dS.m$^{-1}$). In contrast, the high, very high and extreme ($40 \leq$ EC-$_{Lab} \leq 700$ dS.m$^{-1}$) classes are estimated in a range of 30

and 45 dS.m$^{-1}$. It is apparent from the results that model 1 exhibited poor performance, with a low R$^2$ of 0.17 and a very high RMSE of 85%. Likewise, model 5 (Figure 9e) provided similar results, showing insignificant fit (R$^2$ of 0.23) and a very high RMSE (75%).

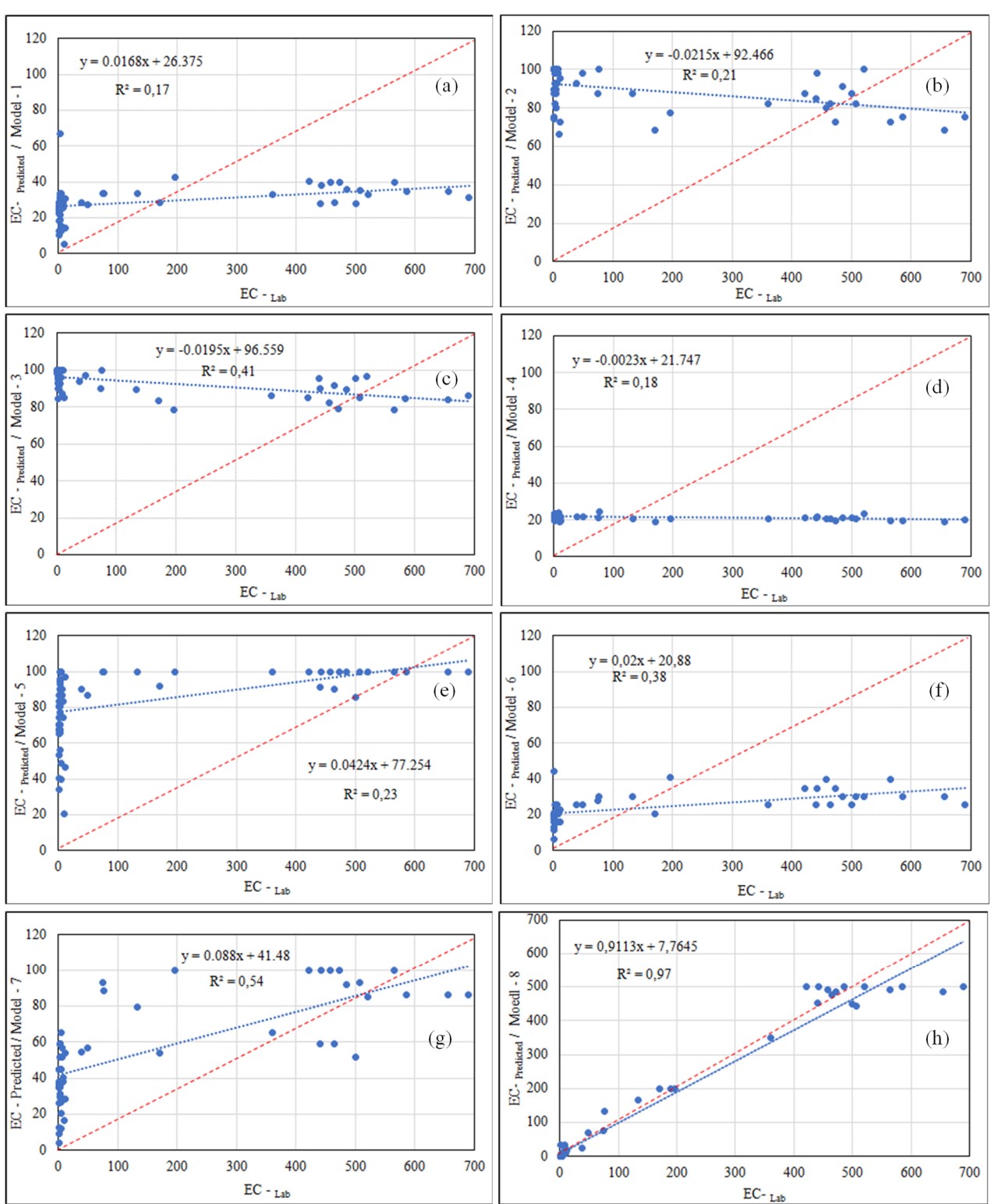

**Figure 9.** Relationship between EC-$_{Lab}$ and EC-$_{Predicted}$ derived from models based on OLI image data.

Integrating the four visible bands of OLI, model 2 predicted the soil salinity classes with insignificant regression fitness (R$^2$ of 0.21) and very high RSME (72%). As illustrated

by Figure 9b, the predicted values vary between 65 and 100 $dS.m^{-1}$ for non-saline soil (EC-$_{Lab}$ $\leq$ 4 $dS.m^{-1}$) as well as for sabkha, which is pure salt (300 to 700 $dS.m^{-1}$). This severe confusion between these classes is in agreement with other relevant studies that have demonstrated the limit of the visible bands for soil salinity classes' discrimination [25]. Comparable behavior and findings were obtained with model 3 (Figure 9c). Confusion occurred for the low salinity or non-saline (EC-$_{Lab}$ $\leq$ 8 $dS.m^{-1}$) classes, which were predicted with a very high range between 65 and 100 $dS.m^{-1}$. A very severe saturation was also observed for high–extreme salinity classes that were underestimated with range values between 80 and 100 $dS.m^{-1}$, while the ground truth observations (EC-$_{Lab}$) were found to range between 100 and 600 $dS.m^{-1}$. Accordingly, the produced regression fit value is low ($R^2$ of 0.41) with a very high RMSE (72%).

Independent to the salt content in the soil samples (1.6 $\leq$ EC-$_{Lab}$ $\leq$ 700 $dS.m^{-1}$), model 4 predicted this variable incorrectly. In fact, the estimated values are very close for all salinity classes, oscillating particularly around 20 $dS.m^{-1}$ (Figure 4d). Based only on the blue band, the regression of this model reveals a very low $R^2$ (0.18) and a very high RMSE (77%). These results corroborate those obtained from the simulation results and visual analysis as discussed above. These results were anticipated, as model 4 considers only the blue band, which is very sensitive to the variation of soil optical properties (color, brightness, texture, etc.) [19,25], and it is particularly dedicated to water resource and coastal zone investigation, as well to the tracking of fine particles (such as those of dust and smoke [72]) in the atmosphere, and not to soil mineralogy detection or mapping.

Proposed for salt-affected soil mapping in vegetated areas at the regional scale, applying a linear logarithmic function and integrating the GDVI, model 6 described the salinity classes with very significant uncertainty levels (Figure 9f). The statistical analysis revealed that $R^2$ is equal to 0.38 with a very strong RMSE (75%). Noticeably, this model showed an overestimate particularly for the non-saline and low salinity classes (1.6 $\leq$ EC-$_{Lab}$ $\leq$ 8 $dS.m^{-1}$) with range values between 10 and 45 $dS.m^{-1}$. Contrariwise, it exhibited an underestimation for the high, very high and extreme (40 $\leq$ EC-$_{Lab}$ $\leq$ 700 $dS.m^{-1}$) salinity classes with a low range between 25 and 40 $dS.m^{-1}$. Developed specifically for soil salinity mapping at the local scale, model 7 is based on a combination of logarithmic and exponential functions by integrating the GDVI and the EM$_H$; it adjusts the observed and predicted values with a relatively acceptable coefficient of determination ($R^2$ of 0.54). However, unfortunately, the calculated RMSE remained very high (65%). This model showed a similar trend to that of model 6, classifying the validation points in two dominant groups (Figure 9g). The first group represents a random mixture of the first five salinity classes (non-saline, low, moderate, high and very high) with overestimation by predicting the range of EC between 3 and 60 $dS.m^{-1}$. Meanwhile, the second group isolated only the extreme salinity class, which was predicted incorrectly in a low range (60 to 100 $dS.m^{-1}$), while the real observed values are between 100 and 700 $dS.m^{-1}$. Contrariwise, model 8 integrates the SWIR bands, which are sensitive to soil–salt mineralogy, and provided the best result of regression fitness, with an $R^2$ of 0.97 and a low RMSE of 0.13. The scatter plot illustrated in Figure 9h reveals a good linear relationship between the EC-$_{Lab}$ and EC-$_{Predicted}$ in terms of good fit 1:1, showing an excellent spatial variability of salinity (1.6 $\leq$ EC-$_{Lab}$ $\leq$ 400 $dSm^{-1}$). While this model was developed for slight and moderate salinity [20], the present study shows appropriate prediction for the different salinity classes investigated.

## 4. Discussion

The results obtained from laboratory analyses revealed that the EC-$_{Lab}$ values are distributed progressively in a wider range between 1.6 and 700 $dS.m^{-1}$ for agricultural fields and sabkha (pure salt), respectively. The results also show that the soil samples are, in general, highly affected by chloride ($Cl^-$), sodium ($Na^+$), magnesium ($Mg^{2+}$) and calcium ($Ca^{2+}$). These major elements are correlated significantly with EC-$_{lab}$, achieving an $R^2$ of 0.98 for chloride ($Cl^-$), 0.87 for sodium ($Na^{2+}$) and 0.48 for $Mg^{2+}$. Moreover, the relation-

ship between $EC_{-Lab}$ and SAR increased progressively and very broadly from non-saline ($EC_{-Lab}$: 1.6 dS.m$^{-1}$, SAR: 0.4) to extreme saline in sabkha ($EC_{-Lab}$: 700 dS.m$^{-1}$, SAR: 445), recording an $R^2$ of 0.70. It was also observed that the measured spectral signatures are automatically linked to salt mineralogy, such as gypsum ($CaSO_4.2H_2O$), halite (NaCl), calcite ($CaCO_3$) and nahcolite (sodium bicarbonate, $NaHCO_3$). Moreover, as illustrated in Figure 4, the spectral signatures of soil samples show different amplitudes and several absorption features depending on the chemical compositions of the existing salts in the selected samples and on other several factors, such as mineralogical composition, salt crystals, impurity, structure, soil texture and the soil optical properties [25,88]. These spectral signatures were acquired in a goniometric laboratory with a very well calibrated ASD, ensuring excellent SNR stability without BRDF problems or atmospheric effects. Of course, the instrument only observes uniform bare soil samples without vegetation cover. This ideal situation optimizes the measured signal, which is attributed only to the bare soil characteristics and salt content in each sample. Hence, these ideal simulation procedure conditions lead to strong validation and comparison among the considered models.

The obtained results based on simulated data show that the first seven models predicted the soil salt content inefficiently with large inaccuracies. Indeed, these models are not able to discriminate between different salt contents in the soil. In general, the six soil salinity classes are predicted, with severe confusion grouping the used 100 soil samples in two major classes only. The first class combines samples of non-saline, low, and moderate salinity classes that are largely overestimated, while the second class includes samples with high, very high and extreme salinity classes, which are unfortunately severely underestimated. Independent of the salt content, their scatter plots show that the 100 sampled points are scattered randomly without any trend or substantial relationship between $EC_{-Lab}$ and $EC_{-Predicted}$ (Figure 7). First-order polynomial regression ($p \leq 0.05$) allowed a poor linear fit to the 1:1 line, yielding a low $R^2$ ($\leq 0.37$) and the highest RMSE ($\leq 275\%$) for models 1, 2, 5, 6 and 7. However, although models 3 and 4 were fitted with $R^2$ of 0.64 and 0.57, respectively, their RMSE remained very large (253%). On the other hand, model 8, integrating the SWIR bands, outperformed the other models, providing satisfactory results. The scatter plot, depicting the relationship between $EC_{-Lab}$ and $EC_{-Predicted}$, clearly illustrates a generally good fit to the 1:1 line (Figure 7h). However, the soil salinity content between 500 and 700 dS.m$^{-1}$ was relatively underestimated, resulting in the data not fitting the 1:1 line very well. This is likely due to either the signal saturation of this extreme class of pure salt, i.e., sabkha salt crust, or the co-crystallization of low solubility (gypsum, calcium and carbonates) and high solubility minerals (NaCl-halite), resulting in low salinities relative to the pure halite salt. Nevertheless, regardless of this minor deviation, this model correctly predicts soil salinity in an arid landscape with an $R^2$ of 0.98 and an RMSE of 5% for non-saline, low and moderate salinity classes, and 10% for the other classes (high, very high, and extreme). According to the results of these simulations, only model 8 provides satisfactory results.

Furthermore, visual analysis of derived maps and their validation by comparison with observations from the field visit, ancillary data and laboratory analysis are consistent with the statistical analyses of the simulation results. In fact, the maps derived using the first seven models (1 to 7) cannot reliably predict the soil salinity classes. Each map shows particular imperfections that are different from each other and, in general, are unable to predict the spatial distribution of soil salinity classes consistently with the ground truth. Thus, due to their low reliability, these maps are not recommended for salt-affected soil prediction in arid landscapes. The established map based on model 8 reveals perfect conformity with the ground truth, effectively predicting the six existing soil salinity classes in the study area. High, very high and extreme salinity classes are located in the coastal zones, lowlands, Bubiyan Island and flat zones of playas. These areas are characterized by very low topography ($\leq 1.2$ m), null slopes and a water table very near to the surface ($\leq 1.00$ m). These physical characteristics promote seawater intrusion and salt accumulation. The richness of this seawater due to soluble salts (sulfates; chlorides; and carbonates of sodium, calcium and magnesium) combined with high temperatures in the region throughout the year accelerate

the evapotranspiration process and, consequently, promote salt accumulation at the soil surface. The moderate salinity class is spatially mapped in the industrial zones and oil field operations. In these zones, sandy and loamy soils are dominant, the terrain elevations are higher than 150 m, and the depth of the water table is greater than 80 m. The low salinity class is mapped correctly in agricultural areas with sandy to loamy soils, low altitude (~20 m) and flat land. Finally, the non-saline class is properly estimated over large areas of flat desert that are located at relatively high topography ($\geq$50 m) and with a deeper water table (>30 m).

In addition to this visual interpretation and validation, the derived maps were statistically validated. The achieved results from this step are comparable to those of the simulations. Indeed, for the first seven models, overestimation can be observed for the non-saline, low and moderate salinity classes, while sever underestimation is noted for the high, very high and extreme salinity classes. Models 1 to 6 exhibited poor performance, yielding an insignificant $R^2$ ($\leq$0.41) and a strong RMSE ($\geq$72%). Unfortunately, although model 7 generated a relatively acceptable $R^2$ (0.54), its RMSE remained high (65%). These first seven models failed to correctly and accurately predict soil salinity classes in an arid landscape. These results are in agreement with those obtained by simulations and corroborate the visual analysis of the derived maps as discussed below. The reason behind this failure resides in the integration of VNIR bands and vegetation indices (NDVI, COSRI and GDVI) in the concepts of these models. Several studies based on field, laboratory and real satellite data acquired with several sensors (TM, ETM+, OLI, ALI EO-1, Sentinel-MSI andWorldView-3) revealed that the VNIR spectral domain lacks the required sensitivity for accurate soil salinity discrimination and quantification. In such wavelengths (coastal, blue, green, red and NIR), the main factors affecting the soil spectral signatures are the salt types, soil mineralogy, level of moisture, organic matter content, color and brightness, roughness and vegetation cover. Undoubtedly, these factors influence the signal gathered by the sensor in a specific pixel size, causing severe confusion between the salt crust in the soils and the intrinsic soil optical properties [17,19,25,88].

Furthermore, due to the SWIR bands, model 8 provided the best regression fitness, with an $R^2$ of 0.97 and a low RMSE of 0.13. However, although these are excellent results, slight overestimation can be observed for the high to very high salinity classes with the $EC_{-Lab}$ range of from 50 to 170 dS.m$^{-1}$, and less underestimation (or saturation) was noted for the extreme salinity classes ($EC_{-Lab} \geq 400$ dS.m$^{-1}$), resulting in the points not fitting the 1:1 line very well. The slope (0.91) and the intercept (7.76) corroborate these observations by expressing a slight deviation from the 1:1 line (Figure 9h). This underestimation is likely attributed to a combination of several problems; for instance, the uncertainty of the measured $EC_{-Lab}$, particularly, for the very high and extreme salinity classes, as well as the impact of soil moisture in sabkha and shorelines (very high and extreme salinity classes) on the signal recorded by the sensor in SWIR bands, which are considerably sensitive to moisture. This may also be due to the saturation of the signal gathered by the sensor over pixels with very high and extreme salinity, i.e., white salt crust, which may be of pure highly soluble halite or a mixture with low-solubility minerals (gypsum, calcite and nahcolite). Alternatively, it could be due to the scale factor that was calculated between field samples identified from an area of about 50 × 50 cm and its homologous points in the OLI image represented by a pixel size of 900 m$^2$. Indeed, the situation is more complicated in the real world when using a multispectral broadband sensor, because variations in the reflectance of soil cannot be attributed to a single salt type or a unique soil property within the individual OLI pixel. In addition, it is also possible that during the preprocessing steps, residual errors persist because each spectral band is corrected uniformly, not pixel by pixel, causing a small difference between the $EC_{-Lab}$ measured at the laboratory, and the homologous $EC_{-Predicted}$ calculated from remote-sensing-derived maps. Eventually, despite these small variations, the validation of model 8 provided satisfactory results in comparison to the other models. These triple validation and comparison procedures based on visual interpretation (by comparison with the field truth and ancillary data) and statistical analyses (similarly

undertaken on simulated and image processing results) highlighted the limit of each model separately and confirmed the potential of model 8 for salt-affected soil mapping in an arid landscape. These results corroborate the findings of relevant studies on irrigated agricultural lands in semi-arid and arid environments [20,32,82].

## 5. Conclusions

This study compared and validated the performance of eight models for soil salinity mapping in an arid landscape using spectral reflectance measurements and a Landsat-OLI image. The examined models were developed for different semi-arid and arid geographic regions around the world. Their modeling concepts consider different spectral bands and are different to mathematical functions. To achieve these goals, a field survey was organized, and 100 topsoil samples were collected with various degrees of salinity classes for spectral reflectance measurements in a goniometric laboratory using an ASD spectro-radiometer. The measured spectra were resampled and convolved in the solar-reflective bands of the OLI sensor using the CAM5S RTC and the filters of the OLI sensor. Thereafter, they were converted in terms of the eight examined models. Additionally, the Landsat-OLI image, acquired at the same time that the field survey was conducted, was radiometrically preprocessed; soil salinity models were implemented; and the soil salinity maps were derived. For validation and comparison purposes, the laboratory analyses were performed to derive $EC_{\text{-Lab}}$ from each soil sample. These steps were undertaken between the predicted $EC_{\text{-Predicted}}$ and the measured $EC_{\text{-Lab}}$ in the same way for simulated and image data using regression analysis ($p < 0.05$). Likewise, the derived maps were visually interpreted and validated by comparison with observations from the field visit, ancillary data (soil, geology, geomorphology and water table maps) and soil laboratory analyses.

The obtained results indicate that independently of data sources (simulated or image) or the validation procedure (statistical or visual), the models based on VNIR spectral bands and/or vegetation indices (NDVI, COSRI and GDVI) are unsuccessful for soil salinity prediction in an arid landscape. This is due to serious signals confusion between the salt crust and soil optical properties in these spectral bands. The statistical tests between observed and predicted salinity values from simulated data or the OLI image converge towards the same results, revealing insignificant fits ($R^2 \leq 0.41$) with very high prediction errors (RMSE $\geq 0.65$). In contrast, the model based on the second-order polynomial function and integrating the SWIR bands provided the results of best fit with the field observations ($EC_{\text{-Lab}}$), yielding an $R^2$ of 0.97 and a low overall RMSE of 0.13. These findings were corroborated by visual interpretation of derived maps and their validation by comparison with the ground truthing and ancillary data.

**Author Contributions:** Z.M.A.-A. and A.B. performed the paper conceptualization, data collection, data pre-processing and processing, results analyses and paper writing. H.R. assisted in the paper writing. A.E.-B. and N.H. participated in field soil survey and spectroradiometric measurements. S.A.S. contributed to soil laboratory analyses and the paper revision. All authors have read and agreed to the published version of the manuscript.

**Funding:** This research was funded by the Arabian Gulf University (Manama, Kingdom of Bahrain) and the International Center for Biosaline Agriculture (Dubai–Emirates). Grant accorded to Professor A. Bannari for the project: Soil Salinity and Properties Mapping in Arid Land Using Remote Sensing, Geographical Information System, Laboratory Analysis, and Field Validation.

**Institutional Review Board Statement:** Not applicable.

**Informed Consent Statement:** Not applicable.

**Data Availability Statement:** The data used to support the results of this research are available on request from the corresponding.

**Acknowledgments:** The authors would like to thank the Arabian Gulf University and the International Center for Biosaline Agriculture for the financial support, as well as the Kuwait government for the scholarship granted to Z.M.A.-A. for her studies. We would also like to thank the NASA-USGS for providing the Landsat datasets and extend our gratitude to the anonymous reviewers for their constructive comments.

**Conflicts of Interest:** The authors declare no conflict of interest.

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
