# Peer review of "Validation and Comparison of Physical Models for Soil Salinity Mapping over an Arid Landscape Using Spectral Reflectance Measurements and Landsat-OLI Data"

_remotesensing, doi:10.3390/rs13030494_

Round 1

Reviewer 1 Report

remotesensing-1078197

The manuscript “Validation and Comparison of Physical Models for Soil Salinity Mapping over Arid Landscape Using Spectral Reflectance Measurements and Landsat-OLI data” addresses an interesting and up-to-date subject, which adhere to Remote Sensing journal policies.

In this research there was compared and validated the performance of eight model of soil salinity mapping in arid landscape using spectral reflectance measurements and Landsat-OLI image.

The manuscript contains interesting results, and presents a good remote sensing application. In addition, the work is well conceived and written, with good figures and discussions, so that I did not identify deficiencies or shortcomings that would require major revisions or improvements.

In my opinion the manuscript must be improved in terms of writing, as some sentences seam unfinished (eg. R20-21), as well as the quality of some of the figures.

Author Response

Reviewer # 1

In my opinion the manuscript must be improved in terms of writing, as some sentences seam unfinished (eg. R20-21), as well as the quality of some of the figures.

Done, sentences are completed and the paper was revised, and linguistic errors were corrected by our department of English language. As well as, the quality of Figure 8 was improved.

Many thanks for your time and your contribution, we appreciate your comments and corrections to improve quality of this paper.

Reviewer 2 Report

The paper deals with an interesting, topical and important topic to lead to the development of knowledge in the field of using spectral reflectance measurements. The paper is well documented and presents reproducible data.

I recommend minor improvements:
- explain / do not use the abbreviations used in the abstract.
- present more explicitly the method for determining soil analyzes in the laboratory.

Author Response

Reviewer # 2

- explain / do not use the abbreviations used in the abstract.

Done

- Present more explicitly the method for determining soil analyzes in the laboratory.

Done.

Many thanks for your contribution; we appreciate your comments and corrections to improve quality of this paper.

Reviewer 3 Report

Dear authors,

I have read your manuscript and in principle I don't have any major objections to the design, methodology, and results. However, I have the impression that the amount of text is disproportional to the number of Figures. In other words, I think the current manuscript size (~10,000 words) can be significantly reduced, without losing the core message of the paper.

Therefore I recommend major revisions.

Some additional minor and editorial comments:

  • L. 117: Sentinel-MSI --> Sentinel-2/MSI
  • L. 131: can you include a short expression on the accuracy of the CAM5S model?
  • L. 156: 46.8 C --> is this the mean maximum temperature? Please specify.
  • L. 275: In --> On
  • L. 329: CORSI --> COSRI
  • L. 415: note --> not
  • Table 6: pHs --> pH
  • L. 791: R(0.41≥), I assume this should be (≤0.41)?
  • L. 840: This study compared and validated

I further advise to have the revised manuscript checked on spelling and grammar by someone with thorough English knowledge.

Author Response

Please, see the attached fiel.

Round 2

Reviewer 3 Report

Dear authors,

I have read your replies and understand that significantly reducing the amount of words would only be possible at the expense of reduced manuscript quality. For me, your justification is sufficient.

The additional minor concerns have been properly addressed, so I want to congratulate you on having an accepted Remote Sensing paper!

This manuscript is a resubmission of an earlier submission. The following is a list of the peer review reports and author responses from that submission.